# Accuracy of potential diagnostic indicators for coeliac disease: a systematic review protocol

Martha Maria Christine Elwenspoek ![ORCID] ,[1,2] Joni Jackson,[1,2] Sarah Dawson,[1,2] Hazel Everitt,[3] Peter Gillett,[4] Alastair D Hay,[2] Hayley E Jones,[2] Deborah L Lane,[5] Susan Mallett,[6] Gerry Robins,[7] Athena Louise Sheppard ![ORCID] ,[8] Jo Stubbs,[5] Howard Thom,[2] Jessica Watson ![ORCID] ,[2] Penny Whiting[2]

For numbered affiliations see end of article.

**Correspondence to**
Dr Martha Maria Christine Elwenspoek;
martha.elwenspoek@bristol.ac.uk

## ABSTRACT

**Introduction** Coeliac disease (CD) is a systemic immune-mediated disorder triggered by gluten in genetically predisposed individuals. CD is diagnosed using a combination of serology tests and endoscopic biopsy of the small intestine. However, because of non-specific symptoms and heterogeneous clinical presentation, diagnosing CD is challenging. Early detection of CD through improved case-finding strategies can improve the response to a gluten-free diet, patients' quality of life and potentially reduce the risk of complications. However, there is a lack of consensus in which groups may benefit from active case-finding.

**Methods and analysis** We will perform a systematic review to determine the accuracy of diagnostic indicators (such as symptoms and risk factors) for CD in adults and children, and thus can help identify patients who should be offered CD testing. MEDLINE, Embase, Cochrane Library and Web of Science will be searched from 1997 until 2020. Screening will be performed in duplicate. Data extraction will be performed by one and checked by a second reviewer. Disagreements will be resolved through discussion or referral to a third reviewer. We will produce a narrative summary of identified prediction models. Studies, where 2×2 data can be extracted or reconstructed, will be treated as diagnostic accuracy studies, that is, the diagnostic indicators are the index tests and CD serology and/or biopsy is the reference standard. For each diagnostic indicator, we will perform a bivariate random-effects meta-analysis of the sensitivity and specificity.

**Ethics and dissemination** Results will be reported in peer-reviewed journals, academic and public presentations and social media. We will convene an implementation panel to advise on the optimum strategy for enhanced dissemination. We will discuss findings with Coeliac UK to help with dissemination to patients. Ethical approval is not applicable, as this is a systematic review and no research participants will be involved.

**PROSPERO registration number** CRD42020170766.

## INTRODUCTION

Coeliac disease (CD) is a systemic, immune-mediated disorder triggered by the ingestion of the protein gluten and related prolamins, found in wheat, rye and barley, in genetically predisposed individuals.[1 2] CD is characterised by small intestine damage, gluten-dependent clinical symptoms and increased blood levels of CD-specific antibodies.[1] Patients typically present with gastrointestinal symptoms, such as chronic diarrhoea, abdominal pain, or constipation, but can also present with non-specific symptoms such as tiredness, anaemia, or impaired growth in children.[3] In the UK and other European countries, the lifetime prevalence is estimated at around 1%.[4–7] CD is more frequently diagnosed in 'at-risk' individuals with certain conditions, such as type 1 diabetes,[8] Down's syndrome,[9] or iron deficiency anaemia,[10] and is 1.5–2 times more common in women than in men.[11]

In first-degree relatives CD prevalence is as high as 10%[12] and concordance in identical twins is 75% higher compared with non-identical twins,[13] suggesting a strong genetic component. Certain human

## Strengths and limitations of this study

► This systematic review protocol follows the Preferred Reporting Items for Systematic Reviews and Meta-Analyses Protocols guidelines, which offers transparency and enhances reproducibility.

► This review will follow guidelines from the Centre for Reviews and Dissemination and the Cochrane Handbook for Systematic Reviews of Diagnostic Test Accuracy, ensuring the use of robust and well-established methodologies.

► We will include studies that use serology tests and/or biopsy as reference a standard to increase the inclusion rate, but we note that serology tests, in particular, can produce both false positive and false negative results (ie, are not a 'gold standard'), which may lead to bias in the primary analysis.

► By balancing the sensitivity and specificity of our search strategy, there is a small risk of missing relevant papers, although this will be mitigated by screening reference lists of relevant reviews.

leucocyte antigen (HLA) variants are strongly associated with CD and almost all patients with CD carry the HLA-DQ2/8 haplotype. However, this risk haplotype is common; up to 40% of the general population carry the HLA-DQ2/8 haplotype—most of whom will never develop CD.[14] Although environmental factors such as gluten exposure, breastfeeding and gastrointestinal infections are reported to play a role in the aetiology of CD,[15] the exact mechanism and pathophysiology of CD remain unclear.

CD is diagnosed using a combination of serology tests and endoscopic biopsy of the small intestine. These tests are only reliable when the patient is on a gluten-containing diet. Current guidelines by the National Institute for Health and Care Excellence (NICE) recommend testing for total immunoglobulin A (IgA) and IgA tissue transglutaminase (tTG) as a first choice in both adults and children.[16] In case of IgA deficiency, IgG tTG, IgG endomysial antibody (EMA) or IgG deamidated gliadin peptide (DGP) can be measured. Seropositive adults should be referred for intestinal biopsy. Seropositive children should be referred for further investigation, which may include IgA EMA, intestinal biopsy, HLA genetic testing or a combination thereof.[16]

The European Society for Paediatric Gastroenterology Hepatology and Nutrition (EPSGHAN) launched new guidelines in 2012[17] to avoid biopsies in children with high IgA tTG titres (>10 times upper limit of normal) and positive HLA-DQ2/8 and EMA results. Since then, new findings have supported this approach.[18] The updated EPSGHAN guidelines (2019) suggest that HLA-DQ2/8 testing can be omitted and the biopsy avoidance strategy can also be used in asymptomatic patients.[18] The biopsy avoidance strategy has been endorsed by the British Society of Paediatric Gastroenterology Hepatology and Nutrition and Coeliac UK (2013) joint Guidelines for the diagnosis and management of CD in children,[4] but not yet by the NICE guidelines.[3] For adults, the biopsy avoidance strategy is currently not recommended.[3 19] However, a recent study suggests that adult CD can also be reliably diagnosed without biopsy in patients with high IgA tTG levels (>10×the upper limit of normal), positive EMA and HLA-DQ2/8 test results without the requirement of symptoms.[20]

Patients diagnosed with CD are advised to follow a strict gluten-free diet, which is expensive and challenging to adhere to for patients, but it is the only available treatment. In most symptomatic patients, a gluten-free diet reduces symptoms, increases the quality of life and seems to lower the risk of complications.[21–23] However, because of the non-specific symptoms and heterogeneous clinical presentation, diagnosing CD is challenging. As a consequence, the mean duration of symptoms before diagnosis is as long as 13 years in the UK[24] and 11 years in the US.[25] A delayed diagnosis can increase the risk of more severe or non-responsive CD,[26] as well as the risk of severe complications, such as osteoporosis,[27] infertility[28] and lymphoma.[29]

Active case-finding strategies, that is, offering tests to individuals with certain symptoms or conditions associated with CD, aim to improve the timeliness of CD diagnosis. Early detection of CD through improved case-finding strategies can improve the response to a gluten-free diet, patients' quality of life and potentially reduce the risk of complications. However, there is a lack of consensus in which groups may benefit from active case-finding. Therefore, we will perform a systematic review of the literature to investigate which diagnostic indicators (such as symptoms and risk factors) are most sensitive and specific for CD, and thus can help identify patients who should be offered CD testing.

## Aim
The overall aim of this review is to determine the accuracy of diagnostic indicators for CD in adults and children.

## Objectives
1. Determine the accuracy of diagnostic indicators to diagnose CD in children (≤16 years old).
2. Determine the accuracy of diagnostic indicators in adults (>16 years old).
3. Identify combinations of diagnostic indicators that lead to an increased probability of CD.

## METHODS
The review will follow Centre for Reviews and Dissemination guidelines for undertaking systematic reviews[30] and methods described in the Cochrane Handbook for Systematic Reviews of Diagnostic Test Accuracy[31] and has been registered on the PROSPERO database. This protocol is reported according to the Preferred Reporting Items for Systematic Reviews and Meta-Analyses (PRISMA) guidance for systematic review protocols and the PRISMA statement for diagnostic test accuracy studies.[32 33] Protocol amendments will be documented and published together with the results of the systematic review.

## Eligibility criteria
Studies that fulfil the following criteria will be eligible for inclusion:
► *Population*: Adults and/or children (≤16 years old) tested for CD.
► *Diagnostic indicators (treated as 'index tests')*: Any diagnostic indicator will be eligible for inclusion. We consider diagnostic indicators to be those that may assist a clinician in making a diagnosis during an initial consultation, for example: family history or recent test results (eg, full blood count to test for anaemia), current symptoms or a risk HLA-DQ genotype. Factors that are more general risk factors rather than potential diagnostic indicators such as perinatal factors, 'susceptibility genes' (other than HLA-DQ status), or age at gluten introduction will be excluded. Tests for susceptibility genes are currently not widely available to clinicians and therefore not (yet) useful in aiding diagnosis.

► *Reference standards*: The following reference standards are acceptable, (1) all patients undergo coeliac-specific serological tests and duodenal biopsy, (2) all patients undergo coeliac specific serological tests and seropositive patients undergo a duodenal biopsy, (3) all patients undergo duodenal biopsy without serology tests and (4) patients undergo one or a combination of the following coeliac specific serology tests: IgA/IgG tTG, EMA or DGP. Studies, where not all patients have received the reference standard, will be excluded.

► *Study design*: Studies using a 'single-gate' or 'multi-gate' design and prediction modelling studies are eligible for inclusion. In a single-gate design, a single set of inclusion criteria (gate) is used for all participants, whereas in a multi-gate design, two or more sets of inclusion criteria (gates) are used (eg, one for a risk factor group or a group of patients with CD and another for healthy controls).[34] Control groups, that is, study participants without the diagnostic indicator or without CD, are required to be representative of the general population. Diagnostic indicator studies will be treated as diagnostic test accuracy studies and we will refer to these studies as such throughout this protocol. If sufficient single-gate studies are identified, then multi-gate studies will be excluded, as the latter have been shown to overestimate test accuracy.[35] Diagnostic accuracy studies will be required to report sufficient data to construct 2×2 tables showing the number of participants with and without CD (according to the reference standard) cross-classified against the number with and without the diagnostic indicator. There will be no specific data requirements for prediction modelling studies.

► *Additional restrictions*: We will exclude studies published before 1997 (the year in which the most important serological test, tTG, was developed), to reduce the variation in the methods used for diagnosing CD (reference standard). We will not apply any restrictions based on patient age, language or study size.

### Information sources
MEDLINE, Embase, Cochrane Library and Web of Science will be searched. Ongoing and completed studies will be identified using the WHO International Clinical Trials Registry and the NIH Clinical Trials database, and internet searches will be undertaken. Reference lists of the latest guidelines from NICE and the British Society of Gastroenterology on CD and systematic reviews published after the latest NICE guidelines will be screened.

### Search strategy
The search strategy incorporates three main elements: (1) conditions (CD)+prognostic/predictive research filter (based on Geersing[36] and Ingui's[37] work), (2) conditions (CD)+all physical diseases/ signs/symptoms (based on MeSH, EMTREE)+'CD' diagnosis and (3) terms for high risk populations (see online supplemental material for a detailed search strategy). We will exclude animal studies, case reports, letters, editorials and coeliac artery/trunk research and will apply a sensitive study design filter. No language or publication restrictions will be applied.

### Study records
#### Data management
Identified references will be downloaded into EndNote V.X9 software for further assessment and handling. Rigorous records are maintained as part of the searching process.

#### Selection process
Study selection will be conducted in two stages using forms developed in Microsoft Access: (1) an initial screening of titles and abstracts against the inclusion criteria to identify potentially relevant papers and (2) screening of the full papers identified as possibly relevant in the initial screening. All papers excluded at the second stage will be documented along with the reasons for exclusion. Abstracts and full texts will be screened independently by at least two researchers. Disagreements about study eligibility will be resolved through discussion or by consulting a third reviewer.

#### Data collection process (data extraction)
Data will be extracted using standardised data extraction forms developed in Microsoft Access. Data extraction forms will be piloted on a small sample of papers and adapted as necessary. In order to minimise bias and errors, data extraction will be performed by one reviewer and checked by a second. Disagreements will be resolved through discussion or referral to a third reviewer.

To exclude diagnostic indicators with insufficient evidence, we will only extract data on diagnostic indicators that are reported in five or more studies, with the exception of 'new but exceptionally promising' diagnostic indicators according to our expert panel (ie, widely used diagnostic indicators that are deemed too important for omission). We will extract the following data from the remaining papers, where reported: study characteristics, patient characteristics, details on the diagnostic indicator and details on how CD was diagnosed (ie, the reference standard).

#### Diagnostic accuracy studies
Numerical results will be extracted as 2×2 contingency tables (ie, number of individuals with vs without the diagnostic indicator, tabulated against CD status). Where possible, data will be extracted separately for different age groups (children and adults), sex (male and female) and symptomatic groups ('at risk' asymptomatic, symptomatic primary care or symptomatic secondary care). If 2×2 data is not reported, we will reconstruct this from estimates of sensitivity, specificity, positive predictive value (PPV) and negative predictive value (NPV), or unadjusted diagnostic ORs, intercept and prevalence data where possible. If this is not possible, the study will be excluded. We will contact authors if there are inconsistencies in the data.

### Prediction modelling studies

We will extract details from the final model, including which variables were used, whether the model was validated, and measures of model performance. If 2×2 data (or sufficient data from which to recover this, such as unadjusted ORs, intercepts and prevalence) are reported for individual diagnostic indicators, these will also be extracted.

### Risk of bias in individual studies

As we are treating diagnostic indicator studies as diagnostic test accuracy studies, risk of bias will be assessed using the QUADAS-2 tool[38] or QUADAS-3 tool if available (currently under development), which includes domains covering participants, index test, reference standard and flow and timing. Prediction modelling studies will be assessed using the PROBAST (Prediction model Risk Of Bias Assessment Tool) tool,[39] which includes domains covering participant selection, outcome, predictors, sample size and flow, and analysis. The content of the tools will be tailored to the review by adding guidance for interpreting signalling questions as appropriate. The tailored tools will be combined and piloted on a small number of studies by two reviewers and agreement assessed. If the agreement is low, the tool and/or guidelines will be further refined until a satisfactory agreement is reached and the tool can be applied to all included studies. If at least one of the domains is rated as 'high risk' the study will be considered at high risk of bias, if all domains are judged as 'low risk' the trial will be considered at low risk of bias, otherwise the trial will be considered at 'unclear' risk of bias. The risk of bias assessment will be conducted as part of the data extraction process.

### Data synthesis

A narrative summary of the included studies will be presented. This will include a summary of the characteristics (eg, study design, population size, geographical location, year, baseline population characteristics, diagnostic indicators evaluated, and reference standard). A detailed commentary on the major methodological problems or biases that affected the studies will also be included.

The analysis will be stratified by age group (children≤16 years old, adults, mixed) and we will group study results according to the type of diagnostic indicator (symptoms, risk conditions, genetic risk factors). For each group, we will plot study-specific estimates of sensitivity and specificity in receiver operating characteristic space and/or produce coupled forest plots.

To exclude diagnostic indicators with insufficient evidence, we will only perform meta-analyses on diagnostic indicators where 2×2 data are available from five studies or more. We will perform a bivariate random-effects meta-analysis of the sensitivity and specificity on each remaining diagnostic indicator.[40 41] This will produce summary estimates of the sensitivity and specificity of each diagnostic indicator, with 95% confidence or credible ellipses, and 95% prediction ellipses. Prediction ellipses

are visual representations of the amount of between-study heterogeneity in both dimensions of a diagnostic test meta-analysis. We will also report estimates of the between-study SD in sensitivity and specificity on the logit scale.

Summary results from each meta-analysis will also be used to estimate PPV and NPV, that is, the probability of CD given that the individual has (or does not have) each diagnostic indicator. Calculation of these values will require an assumed value for the prevalence of CD in the general population. This could be based on studies in the meta-analysis, or on other evidence. If estimates of prevalence are heterogeneous, we will present data or plot predictive values across a range of plausible prevalences.

### Exploration of heterogeneity

We expect heterogeneity in study estimates of sensitivity and specificity due to variability in patient populations, reference standard definitions or methods, study design and other study characteristics.

If sufficient data are available, we will use subgroup analysis and/or meta-regression to investigate whether the identified diagnostic indicators differ in accuracy between: (1) setting (asymptomatic 'at risk', primary care symptomatic, and secondary care symptomatic patients), (2) children versus adults (3) CD diagnosis (biopsy and serology vs serology only), (4) study design (single-gate vs multi-gate), (5) men versus women and (6) low versus the high risk of bias. We will also explore the possibility of adjusting for the imperfect accuracy of the serological tests in a Bayesian statistical framework, using informative prior distributions for the sensitivity and specificity of these tests based on an ongoing systematic review on the accuracy of serology tests for CD.[42]

All statistical analyses will be performed in R and WinBUGS (V.14).

### Prediction modelling studies

We will also produce a narrative summary of prediction models, describing which variables were included in the final model, measures of model performance, and whether the model was validated.

### Patient and public involvement

Our team includes two patient representatives who were involved in the development of the research project, of which this review is a part. The patient representatives will be invited to join regular team meetings to provide patient perspectives on the research to ensure that it addresses patient needs. They will also be involved in the interpretation and dissemination of the results from this review.

### Ethics and dissemination

Results will be reported in peer-reviewed journals, academic, and public presentations and social media. Results will also be expressed in terms of hypothetical populations of 1000 patients for a plain-language summary of analysis results. We will convene an implementation

| Table 1 | Project timeline |
|---|---|
| **Task** | **Deadline** |
| Protocol development | February 2020 |
| Searches | February 2020 |
| Abstract and full-text screening | April 2020 |
| Data extraction | September 2020 |
| Analysis | January 2021 |
| Manuscript submission | April 2021 |

panel to advise on the optimum strategy for enhanced dissemination. We will discuss findings with the charity Coeliac UK to help with dissemination to patients.

## Timeline

This systematic review will be finalised by April 2021 (see table 1 for a detailed timeline of the project).

## DISCUSSION

This systematic review is part of a larger project that aims to identify cost-effective strategies for active case-finding of CD in different at-risk populations. The information identified in the review will feed into economic models to determine the cost-effectiveness of active case-finding strategies, which will help commissioners, clinicians and patients make evidence-based decisions on whether active case-finding for CD should be undertaken in primary care. There is evidence that CD is under-diagnosed in both children and adults.[6 43] Appropriate identification and treatment of CD can have significant benefits for patients in terms of symptoms, quality of life and long-term health outcomes, as well as reducing healthcare and societal economic costs.[24 44] Different guidelines recommend different diagnostic pathways showing a lack of evidence and consensus on which groups may benefit from active case-finding, the best method of doing this, and the costs, benefits and potential harms. A rigorously undertaken high-quality evidence synthesis, including health economic modelling, will provide a robust summary of the current evidence base and highlight whether there is sufficient evidence to suggest an optimum strategy or whether further research is needed.

### Author affiliations
$^1$The National Institute for Health Research Applied Research Collaboration West (NIHR ARC West), University Hospitals Bristol NHS Foundation Trust, Bristol, UK
$^2$Population Health Sciences, Bristol Medical School, University of Bristol, Bristol, UK
$^3$Primary Care, Population Sciences and Medical Education, University of Southampton, Southampton, UK
$^4$Paediatric Gastroenterology, Hepatology and Nutrition Department, Royal Hospital for Sick Children, Edinburgh, UK
$^5$Patient representative, Patient representative, UK
$^6$School of Health and Population Sciences, University of Birmingham, Birmingham, UK
$^7$Department of Gastroenterology, York Teaching Hospital NHS Foundation Trust, York, UK
$^8$Department of Health Sciences, University of Leicester, Leicester, UK

**Contributors** PW conceptualised and designed the protocol. MMCE drafted the initial manuscript. SD, MMCE and PW defined the concepts and search items. PW, MMCE, HEJ and SM planned the data extraction and statistical analysis. HE, PG, ADH, GR and JW provided clinical insights and DLL and JS provided patient perspectives. All authors reviewed the manuscript and have approved and contributed to the final written manuscript.

**Funding** This work was supported by a National Institute for Health Research (NIHR) Health Technology Assessment Programme grant number NIHR129020. This publication presents independent research funded by the NIHR. This research was also supported by the NIHR Applied Research Collaboration West (ARC West). The views expressed in this article are those of the author(s) and not necessarily those of the NIHR or the Department of Health and Social Care.

**Competing interests** None declared.

**Patient and public involvement** Patients and/or the public were involved in the design, or conduct, or reporting, or dissemination plans of this research. Refer to the 'Methods' section for further details.

**Patient consent for publication** Not required.

**Provenance and peer review** Not commissioned; externally peer reviewed.

**ORCID iDs**
Martha Maria Christine Elwenspoek http://orcid.org/0000-0002-9824-9335
Athena Louise Sheppard http://orcid.org/0000-0003-1564-0740
Jessica Watson http://orcid.org/0000-0002-8177-6438

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
