## [Reviewer comments · BMJ Open]

ARTICLE DETAILS

TITLE (PROVISIONAL)	Accuracy of potential diagnostic indicators for coeliac disease: a systematic review protocol
AUTHORS	Elwenspoek, Martha Maria Christine; Jackson, Joni; Dawson, Sarah; Everitt, Hazel; Gillett, Peter; Hay, Alastair D; Jones, Hayley E.; Lane, Deborah L; Mallett, Susan; Robins, Gerry; Sheppard, Athena Louise; Stubbs, Jo; Thom, H; Watson, Jessica; Whiting, Penny

VERSION 1 – REVIEW

REVIEWER	Jonathon Snook Gastroenterology Department Poole Hospital NHSFT Longfleet Road Dorset UK BH15 2JB
REVIEW RETURNED	08-Apr-2020

GENERAL COMMENTS	An interesting and important study proposal. I wish the authors well, and look forward to seeing their results in due course.
---

REVIEWER	Amal Sahyouni Syrian Virtual University Syrian Arab Republic
REVIEW RETURNED	21-Apr-2020

GENERAL COMMENTS	The topic of the research protocol "Accuracy of potential diagnostic indicators for coeliac disease: a systematic review protocol" is interesting and I enjoyed reading this paper.
---

REVIEWER	Michael J. Lentze Dept Pediatrics, University Hospitals Bonn/Germany
REVIEW RETURNED	29-Jul-2020

GENERAL COMMENTS	The protocol for the finding of diagnostic indicators for the diagnosis of coeliac disease is well written. Nevertheless, some minor points should be taken into consideration: 1. Page 6, Line 35: it is correct to look for a control group in children and adults without CD. The control group itself is not well explained (numbers, ages). The reference standard should be better explained.2. Page 8, Line 29: In the protocol children are defined as individuals up to the age of 16. The most accepted definition of a
---

	child is the one of the United Nations Convention on the Rights of the Child, in which a child is an individual under the age of 18. The same definition is true for the UK. So why did you choose ≤ 16 years of age? 3. Page 9, line 6: Apart from assessing the accuracy of serological finding, the accuracy of pathological findings should also be assessed. 4. Page 10: In the discussion the problem of costs is mentioned twice, but the protocol does not give any details as to assessing the costs. Is it in a different protocol? If so, why mentioning it here?
--	---

REVIEWER	Giulia Gibiino AUSL Romagna, Italy
REVIEW RETURNED	07-Aug-2020

GENERAL COMMENTS	The review protocol purposed is an interesting issue on the topic of diagnostic indicators for celiac disease. It will be important to assess conclusive results, in order to be a model applicable in several countries in Europe. I am waiting for final manuscript.
--

VERSION 1 – AUTHOR RESPONSE

Reviewer: 1

Reviewer Name: Jonathon Snook

Institution and Country:

Gastroenterology Department

Poole Hospital NHSFT

Longfleet Road

Dorset

UK BH15 2JB

Competing interests: None declared

Please leave your comments for the authors below

An interesting and important study proposal. I wish the authors well and look forward to seeing their results in due course.

Authors' response

Thank you.

Reviewer: 2

Reviewer Name: Amal Sahyouni

Institution and Country:

Syrian Virtual University

Syrian Arab Republic

Competing interests: None declared

Please leave your comments for the authors below

The topic of the research protocol "Accuracy of potential diagnostic indicators for coeliac disease: a systematic review protocol" is interesting and I enjoyed reading this paper.

Authors' response

Thank you.

Reviewer: 3

Reviewer Name: Michael J. Lentze

Institution and Country: Dept Pediatrics, University Hospitals Bonn/Germany Competing interests: None declared

Please leave your comments for the authors below

The protocol for the finding of diagnostic indicators for the diagnosis of coeliac disease is well written. Nevertheless, some minor points should be taken into consideration:

1. Page 6, Line 35: it is correct to look for a control group in children and adults without CD. The control group itself is not well explained (numbers, ages). The reference standard should be better explained.
2. Page 8, Line 29: In the protocol children are defined as individuals up to the age of 16. The most accepted definition of a child is the one of the United Nations Convention on the Rights of the Child, in which a child is an individual under the age of 18. The same definition is true for the UK. So why did you choose ≤ 16 years of age?
3. Page 9, line 6: Apart from assessing the accuracy of serological finding, the accuracy of pathological findings should also be assessed.
4. Page 10: In the discussion the problem of costs is mentioned twice, but the protocol does not give any details as to assessing the costs. Is it in a different protocol? If so, why mentioning it here?

Authors' response

1. We will include both one-gate and multi-gate designs. Multi-gate designs can either include cases based on coeliac disease or cases based on the diagnostic indicator. The respective control groups will include either healthy participants without coeliac disease or healthy participants without the diagnostic indicator. We will extract a cross tabulation with the number of people with and without coeliac disease and with and without the diagnostic indicator, so both types of multi-gate design can be included in our review. We have changed the explanation of the control group to make this clearer: Page 6, line 24 "In a single-gate design, a single set of inclusion criteria (gate) is used for all participants, whereas in a multi-gate design, two or more sets of inclusion criteria (gates) are used (e.g. one for a risk factor group or a group of coeliac patients and another for healthy controls). Control groups, i.e. study participants without the diagnostic indicator or without CD, are required to be representative of the general population.". We have expanded the explanation of acceptable reference standards: Page 6, line 35: "the following reference standards are acceptable, (1) all patients undergo coeliac specific serological tests and duodenal biopsy, (2) all patients undergo coeliac specific serological tests and seropositive patients undergo a duodenal biopsy, (3) all patients undergo duodenal biopsy without serology tests, (4) patients undergo one or a combination of the following coeliac specific serology tests: IgA/IgG tTG, EMA, or DGP."
2. The cut-off of 16 years was chosen because NICE guidelines for coeliac disease in the UK (NG20) define children as <16 years, young adults as 16-17 years, and adults as 18+ years. The guidelines group young adults and adults together when giving recommendations for diagnosis. In addition, paediatric gastroenterologists tend to take new patients up to age 16, so this cut off was meant to reflect real world practice.
3. We have recently completed a systematic review and meta-analysis of the accuracy of serological tests. This presents us with a unique opportunity to inform robust prior distributions for the sensitivity and specificity of serological tests, allowing the possibility of adjustment in the current review. Unfortunately, it would not be possible for us to do the same for pathology findings, the accuracy of which we have not systematically reviewed. We recognise that biopsy alone is not error-free;

however, the most accurate diagnostic pathway is a combination of serological tests and biopsy results (recommended by UK and EU guidelines), which is much more accurate than serological testing alone. We will, however, carefully acknowledge the limitations of the included reference standards in the study's Limitations.

4. This is in reference to the next step in our research project, namely performing economic modelling to identify cost-effective diagnostic strategies. Costs will not be identified in this review but will be part of the economic analysis. We find it important to mention this because our review sits in a wider research context and its findings will feed into the economic model.

Reviewer: 4

Reviewer Name: Giulia Gibiino

Institution and Country: AUSL Romagna, Italy Competing interests: None declared

Please leave your comments for the authors below

The review protocol purposed is an interesting issue on the topic of diagnostic indicators for celiac disease. It will be important to assess conclusive results, in order to be a model applicable in several countries in Europe. I am waiting for final manuscript.

Authors' response

Thank you.

VERSION 2 – REVIEW

REVIEWER	Michael J. Lentze Dept. Pediatrics. University Hospitals Bonn/German
REVIEW RETURNED	25-Aug-2020
GENERAL COMMENTS	The authors have adressed all the points to my satisfaction